# Angiotensin-I-Converting Enzyme Inhibitory Activity of Protein Hydrolysates Generated from the Macroalga *Laminaria digitata* (Hudson) JV Lamouroux 1813

**DOI:** 10.3390/foods11121792

**Published:** 2022-06-17

**Authors:** Diane Purcell, Michael A. Packer, Maria Hayes

**Affiliations:** 1Food BioSciences, Teagasc, Ashtown, Dublin 15, D15 DY05 Dublin, Ireland; maria.hayes@teagasc.ie; 2Cawthron Institute, 98 Halifax Street, Nelson 7010, New Zealand; mike.packer@cawthron.org.nz

**Keywords:** ACE-1-inhibitory activity, *Laminaria digitata*, seaweed, bioactivity, protein, extract, bioactive peptides, enzyme hydrolysis, antihypertensive activity

## Abstract

Seaweeds have a long history of use as both food and medicine, especially in Asian cultures. Moreover, there is growing interest in the use of seaweed ingredients and bioactive compounds in pharmaceutical and nutraceutical products. One ailment that seaweed bioactive compounds may impact is hypertension caused by the enzyme Angiotensin Converting Enzyme 1 (ACE-1; EC 3.4.15.1), found within the Renin-Angiotensin Aldosterone System (RAAS), which causes vasoconstriction of blood vessels, including veins and arteries. The aim of this paper is to generate bioactive peptide containing protein hydrolysates from the brown seaweed *Laminaria digitata* (Hudson) JV Lamouroux 1813. Proteins were extracted from this seaweed by disrupting the seaweed cell wall using a combination of carbohydrases and proteolytic enzymes. Bioactive peptide containing permeates were generated from *L. digitata* protein hydrolysates, and both hydrolysates and permeates were screened for their ability to inhibit the enzyme ACE-1. The protein content of the permeate fractions was found to be 23.87% compared to the untreated seaweed, which contained 15.08% protein using LECO analysis. Hydrolysis and filtration resulted in a “white” protein powder, and the protein content of this powder increased by 9% compared to the whole seaweed. The total amino acid (TAA) content of the *L. digitata* protein permeate was 53.65 g/100 g of the sample, and contains over 32% essential amino acids (EAA). Furthermore, the *L. digitata* permeate was found to inhibit the ACE-1 enzyme by 75% when compared to the commercial drug Captopril© when assayed at a concentration of 1 mg/mL. The inhibition of ACE-1 (the IC_50_ value) of 590 µg/mL for the *L. digitata* permeate compares well with Captopril©, which had 100% inhibition of ACE-1, with an IC_50_ value of 500 µg/mL. This study indicates that there is potential to develop protein powders with ACE-1 inhibitory bioactivities from the brown seaweed *L. digitata* using enzymatic hydrolysis as a cell disruption and protein extraction/hydrolysate generation procedure.

## 1. Introduction

Brown seaweeds, also known as Kelp, form an important part of the marine ecosystem and are a common food source in Asian society [1]. Kelp are a potential bioresource for the generation of ingredients for use in pharmaceuticals, nutraceuticals, functional foods, bioenergy, textiles and bioplastics.

At present, beyond food and feed uses of seaweed, the main commercial global seaweed products are hydrocolloids, which include alginates, agar and carrageenan. Alginates contribute 31% of the global hydrocolloid demand, and one of the most important phycocolloids is alginate, used as a gelling, thickening and emulsifying agent. The brown seaweed *Laminaria digitata* (*L. digitata*), commonly called Oarweed, is used as a food, feed and a food additive in the form of commercial alginate [2,3,4].

Due to an increased global prevalence of health issues associated with metabolic syndrome, seaweeds are considered a potential resource of bioactive compounds that may prevent or treat several ailments including diabetes, high blood pressure and inflammation [5,6,7,8,9]. Brown seaweeds have now been investigated extensively for their alginate and polyphenol bioactivity, and potential application in pharmaceuticals [10,11,12,13].

Hypertension data collected over the past 20 years found a doubling in the number of adults with hypertension, to reach 626 million men and 652 million women in 2019, which equates to approximately 26% of the global adult population, or 1.28 billion people. This number is set to rise to 29% of the global population by 2025, to reach 1.56 billion people [14,15,16]. Arterial Hypertension (AHT) is ordinarily treated using drugs that inhibit the Angiotensin I-Converting Enzyme (ACE-1; EC 3.4.15.1) such as Enalapril^®^ or Captopril© [17,18,19,20]. However, there are side effects associated with these drugs, and this has prompted the pursuit of natural remedies that may prevent the development of high blood pressure [5,6,21,22]. ACE-1 is a zinc metallic protease, which functions as a catalyst in the conversion of Angiotensin I to a powerful vasoconstrictor Angiotensin II, and increases the degradation of the vasodilator Bradykinin [23]. The benefit of the control of this enzyme includes a decrease in Angiotensin II concentrations and an increase in the vasodilator Bradykinin, the combination of which results in a reduction in hypertension or high blood pressure [24,25,26]. The most frequently studied natural ACE-1-inhibitors are protein hydrolysates containing bioactive, ACE-1 inhibitory peptides derived from different protein sources including dairy, marine and plant proteins [22,27]. Other molecules that may inhibit ACE-1 include polyphenols [6,28,29,30]. Previous studies have shown the ACE-1 and antihypertensive activities of the brown seaweed *Undaria pinnatifida*. When administered to human subjects with high blood pressure as a dietary supplement, this seaweed was found to decrease systolic blood pressure (SBP) by 13 mmHg [31]. Methanol extracts from two brown seaweeds, *Sargassum siliquosum* and *Sargassum polycystum*, previously displayed ACE-1 IC_50_ values of 0.03–0.42 mg/mL, from concentrations of 0.2–1.0 mg/mL [13].

Moreover, phlorotannins from *Sargassum wightii* also displayed ACE-1 inhibitory activity, with a reported IC_50_ value of 56.96 µg/mL. This equates to the IC_50_ value previously reported for Captopril © 51.79 µg/mL [12]. In addition, protein hydrolysates extracted from the green seaweed *Ulva intestinalis* had ACE-1 inhibitory IC_50_ values of 0.183 mg/mL and 0.179 mg/mL when assessed previously [32].

Although one of the world’s most cultivated seaweeds species, *L. digitata* ACE-1 inhibitory protein hydrolysates are not documented in the literature. In this work, we report the ACE-1 inhibitory activity of a protein hydrolysate generated from *L. digitata* using enzyme hydrolysis to disrupt the seaweed cell wall and release small proteins and peptides with ACE-1 inhibitory activity. The seaweed cell wall was broken using the enzymes Viscozyme^®^ and peptides generated using Alcalase^®^ protease from *Bacillus licheniformis*. The nutritional composition and bioactivities of the hydrolysates and permeates generated were determined, as well as the total amino acid (TAA) composition of all fractions. Active permeates that were found to inhibit ACE-1 by greater than 75% were enriched and purified further, using molecular weight cut off (MWCO) filtration. The IC_50_ value for the active permeate fractions was determined and compared to other hydrolysates of marine origin.

## 2. Materials and Methods

### 2.1. Chemicals and Reagents

Hydrochloric acid (HCl), sodium hydroxide (NaOH), Viscozyme^®^ L enzyme and Protease from *Bacillus licheniformis* enzyme were supplied by Merck (formerly Sigma) (Dublin, Ireland). The Angiotensin I-converting enzyme (ACE-1) inhibition assay kit was supplied by Cambridge Scientific (Cambridge, UK); Captopril© was supplied by Sigma Aldrich (Dublin, Ireland). The 3-kDa Millipore Prep/Scale TFF filtration system was supplied by Merck Millipore (Cork, Ireland). Deionized water obtained from a Millipore nano-water purification and filtration system was supplied by Sigma-Aldrich (Dublin, Ireland).

### 2.2. Generation of Peptide Hydrolysates from L. digitata

*L. digitata* (Hudson) JV Lamouroux 1813 was supplied dried and milled by the Irish company SeaLac Ltd, Sligo, Ireland. Seaweed was harvested from the Kinvarra coastline, Galway, Ireland during the spring of 2021, and subsequently air dried, milled and supplied for use in this study. 100 g of dried seaweed was added in triplicate to 500 mL of MilliQ deionized water, and subsequently mixed and heat-treated in a water bath (Stuart, UK) at 95 °C for 15 min to deactivate endogenous enzymes. The mixture was cooled to room temperature, and Viscozyme^®^ was added to the biomass at an enzyme to substrate ratio of 1% (*v*/*w*). Hydrolysis was performed in a water bath at 45 °C for 4 h. The pH was maintained at 4.5 pH using NaOH (2 M), and the amount added to keep the pH at the optimum for Viscozyme^®^ was recorded and used to determine the degree of hydrolysis (DH) using the Adler-Nilsen method [33]. Following hydrolysis with Viscozyme^®^, the enzyme was heat-deactivated at 95°C for 15 min, the hydrolysate was cooled to room temperature and Alcalase^®^ was added to the substrate at a ratio of 1% (*v*/*w*). Hydrolysis was again performed in a water bath with conditions set at 55 °C, pH 7 for 2 h. Again, the amount of base added to keep the pH at 7 was recorded, the enzyme was heat deactivated at 95 °C for 15 min after 2 h of hydrolysis. The hydrolysate was subsequently frozen at −80 °C and freeze-dried using a Cudden Engineering freeze drier (New Zealand) (Figure 1).

### 2.3. Peptide Enrichment Using Molecular Weight Cut Off (MWCO) Filtration

The Millipore lab-scale Tangible Flow Filtration (TFF) system was used in combination with a 3 kDa cellulose filter membrane to generate a peptide enriched permeate fraction and retentate fraction. The membrane was cleaned using 0.1 M NaOH at 45 °C followed by washing with ddH_2_O prior to passing the hydrolysates through the system. Permeates and retentate were frozen and freeze-dried, as described in Section 2.2 for hydrolysates (Figure 1).

### 2.4. Chemical Characterisation of Seaweed and Seaweed Hydrolysates and Permeates

Determination of the total amino acid composition of samples was determined using hydrolysis with 6 M HCl at 110 °C for 23 h, [34]. The samples were then deproteinized by mixing equal volumes of 24% (*w*/*v*) tri-chloroacetic acid and sample. Samples were allowed to stand at room temperature for 10 min before centrifugation at 14,400× *g* for 10 min. The supernatants were recovered and diluted with 0.2 M sodium citrate buffer, pH 2.2, to give approximately 250 nM of each amino acid residue. Samples were then diluted 1:2 with the internal standard nor-leucine to give a final concentration of 125 nM/mL. Amino acids were quantified using a Jeol JLC-500/V amino acid analyser (Jeol Ltd., Garden city, Herts, UK) fitted with a Jeol Na+ high performance cation exchange column.

#### 2.4.1. Protein Content

The protein content of all samples was determined using a LECO FP628 (LECO Corp., MI, USA) protein analyzer based on the Dumas method, which can also be referred to as the combustion method, and was carried out according to AOAC method 992.15, 1995, [35]. Briefly, each sample was weighed into foil cups and the mass recorded before placement into numbered carousels of the LECO FP628 for combustion and nitrogen analysis. A nitrogen conversion factor of 5.0 was used to calculate the concentration of protein in these seaweed samples, since the Dumas method measures total nitrogen [36,37].

#### 2.4.2. Fat and Moisture Content

The fat content of all the samples was determined using the ORACLE Rapid NMR Fat Analyser (CEM Corp., Charlotte, NC, USA). Samples were weighed onto previously weighed CEM glass fiber sample pads and dried in a hot air oven overnight. The weight of the dried samples was calculated (moisture loss), and pads with the dried samples were each wrapped in a single sheet of Trac Film, placed in an ORACLE tube, and heated to 40 °C on a heat block before fat analysis by the ORACLE instrument. This method was validated in accordance with AOAC official method 2008.06, and AOAC PVM 1:2004. The moisture content of the *L. digitata* was determined by oven drying at 105 °C, and the ash was evaluated by sample incineration at 515–540 °C in a muffle furnace for up to 24 h with modifications as described previously [38].

#### 2.4.3. Ash Content

Ash content of all the samples was determined using the Muffle furnace. Samples were weighed in dried and marked crucibles. Before the ashing procedure, samples were initially charred by initiating a pre-burn process on a hotplate for 3 h at 100 °C. The pre-burned samples were carefully transferred to a muffle furnace for ashing overnight at 540 °C. After ashing was complete, samples were cooled to room temperature in a desiccator, and the weight of the percentage ash was calculated using the following formula
(1)$$\%\;\mathrm{Ash}=\frac{M_{2}-\;M_{0}}{M_{1}-M_{0}}$$
where: *M*_0_ = mass in grams of dish; *M*_1_ = mass in grams of dish and test portion; *M*_2_ = mass in grams of dish and ash

Total carbohydrate content was determined by calculation with all other proximate data added and taken from the 100%. All concentrations were expressed as a percentage of algal dry weight (DW).

### 2.5. Degree of Hydrolysis (DH) of L. digitata

DH calculation [33]
(2)$$\mathrm{DH}=B\times N_{B}\times \frac{1}{\alpha }\times \frac{1}{M_{p}}\times \frac{1}{h_{tot}}\times 100\%$$
*B* (mL) = 15 mL
*NB* = 1.5 Moles
1α = 2.27
*Htot* = 8 amino meq/g
*Mp* = 10
$$\mathrm{DH}=((15\times 1.5)\times (2.27)\times (\frac{1}{10})\times (\frac{1}{8}))\times 100=63.84\%$$

### 2.6. Statistical Analysis

All yields and nutrient composition value calculations and bioactivities were expressed as mean values ± standard deviation (SD) (n = 3). The Analysis Tool Pak in Microsoft Excel Office was used. Statistical comparison using ANOVA (one-way), with differences between means at the 5% (*p* < 0.05) level, were considered significant.

### 2.7. ACE-Inhibitory Activity Determination of L. digitata of 3 kDa Protein Hydrolysates, Permeates, and IC_50_

The ACE-I inhibitory activity was determined using a test kit supplied by Cambridge BioSciences (Cambridge, UK) with slight modifications. Briefly, the enzyme working solution and indicator working solution were prepared according to the manufacturer’s instructions. We added 20 µL of a 1% (*w*/*v*) solution of each fraction and 46.02 mM of captopril (a known ACE inhibitor) dissolved in distilled water to the experimental sample wells of a 96-well microplate and 20 µL and 40 µL deionized water for the blank 1 and blank 2 wells, respectively. Then, 20-µL substrate buffer was added to each well, followed by 20-µL enzyme working solution to experimental wells and blank 1. The plate was then incubated at 37 °C for 1 h. After incubation, 200 µL indicator working solution was added to each well, and the plate was incubated at room temperature for 10 min. Absorbance was measured at 450 nm on a microplate autoreader, and ACE-1 inhibitory activity was calculated using the following equation:

Percentage ACE-1 inhibition = (Absorbance at 45 nm of blank1 − Absorbance at 450 nm of sample)/(Absorbance at 450 nm of blank1 − Absorbance at 450 nm of blank 2) × 100 [39,40]. Where; blank 1 is the control without the addition of any inhibitor, blank 2 is the reagent blank, and the inhibitor is the positive control (Captopril©) or test sample (protein hydrolysate fraction).

For IC_50_ determination, the ACE kit-WST was also utilized, and samples were subjected to a set of six serial dilutions (using a sample starting volume of 60 µL), to establish an inhibition curve [39,40].

## 3. Results

### 3.1. Extraction Yields

Figure 2 is an image of the raw whole unrefined seaweed *L. digitata*, the hydrolysate of *L. digitata*, post enzymatic treatment, and the permeate fraction of *L. digitata* recovered post 3-kDa MWCO filtration.

As shown in Figure 3, an increase of 9% protein content was observed for the protein hydrolysate in comparison to the whole untreated seaweed. The lipid and ash content of the hydrolysate and permeate increased by 0.1% and 24%, respectively, compared to the whole, untreated seaweed. Carbohydrate content decreased by 25% in the hydrolysate and moisture decreased by 2.7% in the hydrolysate compared to values shown for whole untreated seaweed. The yield of protein powder recovered during the process is also shown in Figure 3. After the first stage of enzymatic hydrolysis, the total percentage loss in biomass when supernatant (12.68 g) and retentate (79.71 g) (not listed) are added together was only 7.61% or 7.61 g as 100 g of raw seaweed was used as the starting biomass. The final fraction permeate post 3-kDa filtration noted a decrease is yield of 53.11%, indicating that of the 12.68 g supernatant, only 4.01 g, or 4% of the total raw seaweed biomass, was <3 kDa.

### 3.2. Nutritional Composition of Whole L. digitata, Hydrolysates and Permeates

The chemical composition of untreated *L. digitata* is shown in Table 1. The ash content of *L. digitata* was 27.83 ± 2.81% of the dry weight (DW). Protein was 15.08 ± 0.2% of DW. The lipid content of the whole seaweed was 0.5 ± 0.25% of DW, which is in line with values reported previously for brown seaweeds [41,42]. Moisture is normally the highest constituent for all seaweed, at approximately 85% wet weight (WT) [43], and the dry weight of *L. digitata* was 11.2 ± 0.03%. The carbohydrate content of brown seaweeds is between 13–60% of DW, and this *L. digitata* sample contained 41.62 ± 3.28% carbohydrates [44,45].

Table 1 also lists the proximate analysis of the protein hydrolysate (supernatant). Ash content of this sample was 49.11 ± 1.91%, an increase of 9% compared to the ash content of untreated, whole *L. digitata*. The protein content of the hydrolysate was 21.86 ± 0.10% an increase of 6% compared to the content of protein in the raw, untreated *L. digitata*. This indicates that the enzyme extraction method was effective at releasing protein from the sample matrix. The degree of hydrolysis (DH) is 63.84%, which indicates the release of protein from the biomass. The lipid content of the hydrolysate was 0.6 ± 0.28% and moisture was 8.24 ± 0.18%. The carbohydrate content of the hydrolysate was 20.19 ± 2.47% compared to 41.62 ± 3.28% carbohydrate reported for untreated *L. digitata*.

The ash content of the residual biomass is just over half the value observed for the supernatant at 27.84 ± 1.91%. Protein values obtained using the LECO method are similar to values recorded for the untreated *L. digitata* sample at 14.88 ± 0.33%. Lipid content was in a similar range to the supernatant 0.72 ± 0.35%.

The proximate values for the permeates and retentates obtained following 3-kDa filtration were recorded (Table 1). The permeate ash value was 52.26 ± 0.8%. This is due to the concentration of salts using MWCO filtration. Filtration can be used as a method to remove salts and a 3-kDa filter enriched for salt as previously described [46]. The protein content in the permeate fraction was 23.87 ± 0.13%. Enzymatic treatment and filtration increased the protein content by ~9% from the protein value observed for untreated *L. digitata*. Lipid values did not change at 0.6 ± 0.25%. Moisture remained similar to the protein hydrolysate supernatant at 8.50 ± 0.49%. Carbohydrate dropped by ~5% to 14.77 ± 1.67%. Proximate results for the *L. digitata* retentate are also recorded. The retentate ash value was 38.82 ± 1.61%. Ash was enriched in the permeate fraction due to the size of the MWCO filter used. The retentate fraction still contained proteins larger than 3-kDa in size and the percentage protein in the retentate was 18.59 ± 0.18%. The lipid content of the retentate increased by 1% compared to the value observed in the untreated seaweed. The retentate fraction carbohydrate value was double that observed in the permeate fraction.

### 3.3. Total Amino Acid (TAA) Profiles for Raw Seaweed (a), and 3 kDa Hydrolysate Permeate

The amino acid profile of untreated whole *L. digitata* and the permeate fractions were determined. Seventeen amino acids were identified in the untreated *L. digitata*, and the 3-kDa filtered protein hydrolysate permeate (Table 2). The total amino acid (TAA) content of the untreated *L. digitata* was 161.93 g/100 g of sample and *L. digitata* was found to consist of 35.67% essential amino acids (EAA). The most abundant amino acid was alanine in both raw seaweed and 3-kDa permeate fraction with values of (32.78 g/100 g), and the (63.69 g/100 g), respectively (Table 2). Previously, seaweeds *Fucus vesiculosus* and *Alaria esculenta* were identified as containing 12–41% EAA [47]. The four most abundant amino acids in the raw seaweed fraction are non-essential amino acids, in alanine, arginine, aspartic acid, and glutamic acid [40]. For the 3-kDa permeate the top 4 amino acids were slightly different to the raw seaweed where alanine was highest, then proline, aspartic acid and glutamic acid. Proline is a secondary amine, and used in the pharmaceutical industry as an osmoprotectant, which protects organisms from osmotic stress [48]. Interestingly, glutamic acid and aspartic acid are identified as dietary feeding attractants for several fish species [49,50], and alanine is considered a vital energy substrate for fish and a favored carrier for nitrogen for inter-organ amino acid metabolism in fish [51]. Based on this information a potential use for the permeate fraction containing non-essential amino acids is as a dietary supplement used for fish diets.

### 3.4. ACE-1 Inhibition of Seaweed Extracts

The ability of the permeate fraction to inhibit ACE-1 activity was determined following hydrolysis, centrifugation and 3-kDa filtration. The highest percentage inhibition of ACE-1 was observed for the permeate sample where inhibition of ACE-1 was 65–76% ± 5.33 when the permeate was assayed at a concentration of 1 mg/mL compared to the positive control Captopril©.

The ACE-1 IC_50_ value obtained for the *L. digitata* 3-kDa permeate was 590 µg/mL (Table 3). This is a promising result when compared with Captopril© a drug used to control AHT that achieved 100 % inhibition, and an IC_50_ of 500 µg/mL In addition, the IC_50_ value of a protein hydrolysate generated from *Undaria* sp. (Wakame) was shown as 86 µg/mL. This hydrolysate was tested further for antihypertensive activity in an animal trial, where it was orally administered to spontaneously hypertensive rats (SHRs). The hydrolysate was fed orally at concentrations of 10 mg and 100 mg protein/kg body weight (BW), and compared to the positive antihypertensive drug Captopril©, which was administered at a dose of 5 mg/kg BW. Three hours post-administration of 10 mg/kg BW of the hydrolysate, systolic blood pressure (SBP) dropped by 19.3 ± 3.1 mm Hg (*p* < 0.01), and after six hours post-administration of 100 mg protein/kg BW of the hydrolysate, SBP decreased by 23.5 ± 5.3 mm Hg (*p* < 0.05). These in-vivo values compared favorably to Captopril©, which decreased SBP in SHRs by 30.6 ± 10.2 mm Hg (*p* < 0.05), 3 h post-administration [52]. In addition, previously generated hydrolysates from the green seaweed, *Ulva intestinalis* reported ACE-1 inhibition values of 0.183 mg/mL and 0.179 mg/mL when assayed at different concentrations. Observed ACE-1 inhibition was attributed to interaction of Zn^2+^ and the two novel peptides identified and characterized from the hydrolysate [45].

## 4. Discussion

The aim of this work was to use a solvent free extraction method to extract protein from seaweeds, and to reduce costs, minimize biomass loss, improve yield and make the method applicable at scale. The protein extract obtained from *L. digitata* following hydrolysis, centrifugation and MWCO filtration exceeds protein values reported for whole brown seaweeds previously of less than 15% protein content on a dry weight basis [32,43,53,54]. The ash content of the permeate sample was 52.26± 0.81% and this poses some issues for potential application of the permeate in feed or food products. However, the ash content of whole *L. digitata* was reported at 31.6 ± 7.1% DW previously [54], and ash content could be reduced by washing of biomass prior to drying. Lipid values in this study remained within a narrow range, and are typical of lipid contents previously reported for brown seaweeds, which were found to contain between 0.5–5% DW lipids [41,42].

Of the total amino acids detected, over 32% of these were EAAs. Previous work did compare several extraction methods, and noted that for other brown seaweeds including *Fucus vesiculosus* and *Alaria esculenta* used an autoclave method that produced the highest yield of EAAs from *Fucus* sp. of (26.55%) [48].

An increase of protein content from 9% in the raw biomass, to 24% protein in the permeate fraction, was observed in this study. Previous work on *L. digitata* found a protein content of 6.9 ± 1.1% DW for seaweed harvested off the coast of Scotland [54,55]. Use of enzyme-assisted extraction (EAE) can considerably improve extraction yields, as shown in this work. In addition, this method is both economic and scalable. Results obtained here concur with use of EAE in previous studies, where it was used to extract protein from red and green seaweeds [56,57].

ACE-1 is one of the key elements responsible for vasopressor action [23]. ACE-1 converts angiotensin–I to angiotensin-II, a potent vasopressor, in the renin-angiotensin-aldosterone system (RAAS), and contributes to increased blood pressure by inactivating bradykinin, a strong antihypertensive peptide [39,40]. The hydrolysates and permeates generated in this study were both tested for their ability to inhibit ACE-1. The literature regarding the generation of ACE-1 inhibitory peptides, specifically from brown algae, is not extensive. This study found that the lower molecular weight fraction (<3-kDa) from a protein hydrolysate from *L. digitata* inhibited ACE-1 by 75% when assayed at a concentration of 1 mg/mL. This is the first study on *L. digitata* to document this type of enzymatic inhibition linked to protein hydrolysates and bioactive peptides. Previous studies have outlined the potential bioactive benefits of hydrolysates linked specifically to peptides [32]. This includes other brown seaweeds, such as *Ecklonia cava* that noted inhibition of ACE-1 by 90% following an aqueous extraction at 70 °C followed by hydrolysis with the food-grade enzyme, Flavourzyme [55]. Moreover, a 3-kDa fraction generated from another brown seaweed *Fucus spiralis* L. inhibited ACE-1 by 86.85 ± 1.89% [58], and the edible brown seaweed *Undaria pinnatifida* hydrolysates, had antihypertensive activity in spontaneously hypertensive rats (SHR), with the most effective ACE-1 inhibitor fractions found when using protease S ‘Amano’ [59]. In addition, an enzymatic extraction method using α-amylase produced ACE-1 inhibitory activity from three brown seaweeds, and 95.61 ± 0.3% ACE-1 inhibition was observed for a hydrolysate generated from *L. nigrescens* [60]. These studies indicate that several brown seaweeds have bioactivity effective against the ACE-1 enzyme, and could consequently be used to control hypertension in humans. Based on the results from this study, the bioactivity is linked to bioactive peptides generated during the hydrolysis and filtration process within the 3-kDa permeate. Bioactive peptides have several documented biological effects including antihypertensive, antioxidant, antithrombotic, antimicrobial and immunomodulatory properties [60,61,62,63]. In addition, the use of this extraction method, that is solvent-free and scalable, also makes it a potentially economic and viable option to add value to this brown seaweed [64,65,66,67,68,69]. Extraction costs, including use of solvents, are a documented bottleneck in bio-product manufacturing [49,70]. These results build on previous work carried out concerning the health beneficial properties of brown seaweeds linked to polyphenols and carbohydrate fractions. For example, recently Costa and colleagues demonstrated that dietary inclusion of 15% *Laminaria digitata*, supplemented or not with carbohydrases, could improve the nutritional value of poultry meat without impairing animal growth performance [71]. The authors attributed this to polyphenols and bioactive carbohydrates [71]. Previously, Chen the ACE-1 inhibitory activity from a protein hydrolysate generated from *Saccharina (Laminaria*) *japonica* and linked the observed activity to seven bioactive peptides with the amino acid sequences KY, GKY, STKY, AKY, AKYSY, KKFY and KFKY [54]. Moreover, the reported bioactivity of the protein hydrolysates compare favorably with ACE-1 inhibitory peptide hydrolysate values identified previously from dairy whey protein hydrolysates (reported ACE IC_50_ values of 345–1733 µg/mL) [72], meat (beef derived ACE-1 inhibitory peptides had IC_50_ values of 0.89, 2.69, and 3.09 mM, respectively) [73], mussel *Mytilus galloprovincialis* (IC_50_ for the inhibition of ACE of 1.0 to 0.56 mg of protein/mL) [74], microalgal (a few studies have reported IC_50_ values of microalgal ACE-1 inhibitory peptides ranging from 0.5 to 474.36 μM) [75] and seaweed sources such as *Fucus spiralis* [47].

## 5. Conclusions

This work is the first to demonstrate the extraction of proteins and generation of ACE-1 inhibitory hydrolysates from *L. digitata* using a combination of different food grade enzymes. To the best of the authors’ knowledge, this is the first paper to report ACE-1 inhibitory activity for protein hydrolysates generated from *L. digitata.* The next steps in this work will involve characterization of the bioactive peptides responsible for the observed ACE-1 inhibitory activity. *L. digitata* is cultivated globally through aquaculture and this fact, combined with the potential to extract protein and generate bioactives with heart health benefits makes it a suitable candidate for product development. This work adds to the evidence that seaweed-based health maintenance strategies are now considered viable methods to improve global human health.

## Figures and Tables

**Figure 1 foods-11-01792-f001:**
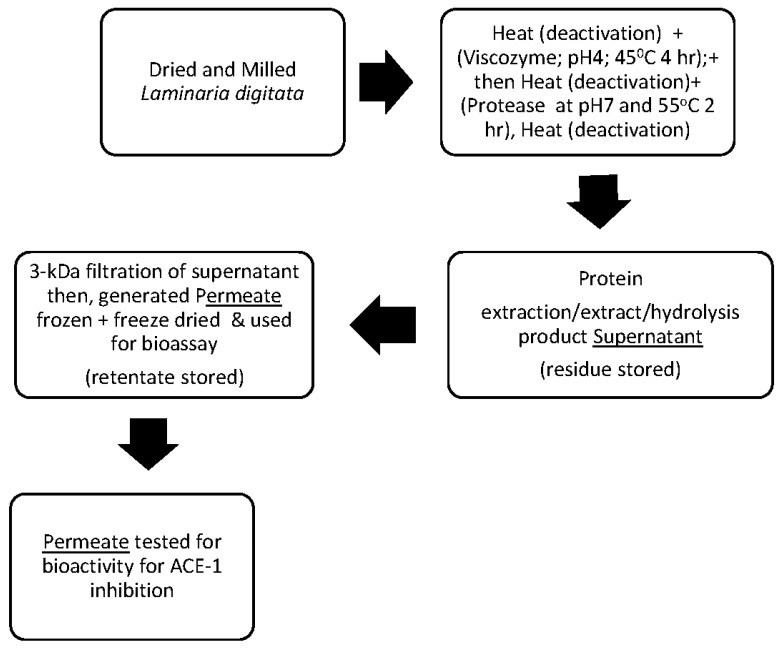
Schematic representation of the process used to generate ACE-1 inhibitory permeates from *L. digitata.* Briefly, milled and dried *L. digitata* was heat deactivated and hydrolysed with two enzymes—Viscozyme^®^ and Alcalase^®^. Hydrolysates were filtered to generate permeate and retentate fractions using a Millipore Lab scale TFF system and a 3 kDa cellulose filter. Fractions were subsequently tested for their ability to inhibit the ACE-1 enzyme using a spectrophotometric method.

**Figure 2 foods-11-01792-f002:**
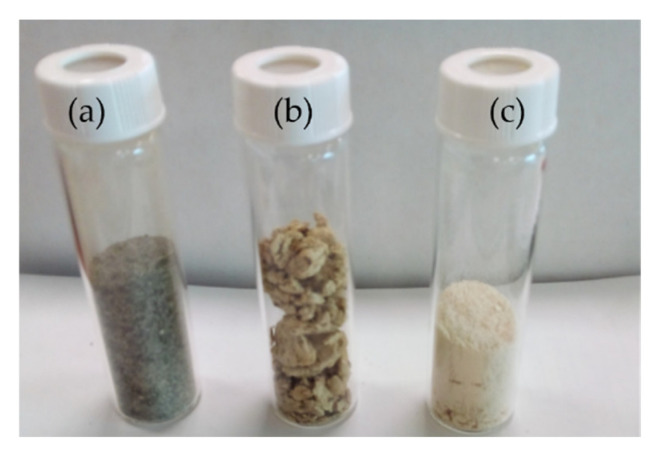
(**a**) *L. digitata* whole seaweed; (**b**) freeze-dried *L. digitata* hydrolysates generated using Viscozyme and the protease enzyme Alcalase^®^ and (**c**) the *L. digitata* permeate fraction generated by filtration with a 3-kDa MWCO filter permeate fraction recovered.

**Figure 3 foods-11-01792-f003:**
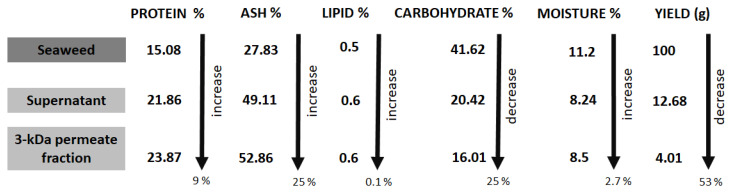
Schematic representation of the proximate composition in percentage for the protein, ash, lipid, carbohydrate, moisture content of fractions generated from *L. digitata* using hydrolysis and centrifugation. Percentage yields (%) are reported for each fraction—the supernatant and the 3-kDa permeate fraction.

**Table 1 foods-11-01792-t001:** Initial compositional analysis values for *L. digitata* whole biomass and fractions generated following enzyme treatment and centrifugation steps and subsequent 3-kDa MWCO filtration steps.

Sample Name	% Protein	SD	% Lipid	SD	% Carbohydrate	SD	% Ash	SD	% Moisture	SD
Whole untreated *L. digitata* (a)	15.08	0.2	0.5	0.25	41.62	3.28	27.83	2.81	11.2	0.03
*L. digitata* supernatant fraction recovered following centrifugation (b)	21.86	0.10	0.60	0.28	20.19	2.47	49.11	1.91	8.24	0.18
*L. digitata* residual biomass fraction following centrifugation (b)	14.88	0.33	0.72	0.35	50.85	2.75	27.84	1.91	5.73	0.16
*L. digitata* permeate fraction recovered following MWCO filtration with a 3-kDa membrane (c)	23.87	0.13	0.60	0.25	14.77	1.67	52.26	0.81	8.50	0.49
*L. digitata* retentate fraction recovered following MWCO filtration with a 3-kDa membrane (c)	18.59	0.18	1.56	0.62	31.13	4.19	38.82	1.61	9.90	1.79

SD = (standard deviation) N = 3.

**Table 2 foods-11-01792-t002:** Total amino acids content of untreated *L. digitata* and *L. digitata* 3-kDa permeate fraction recovered following MWCO filtration.

Amino Acid	*L. digitata* Raw Seaweed (g/100 g)	*L. digitata* 3-kDa Permeate (g/100 g)
Alanine	32.78	63.69
Arginine	7.84	2.44
Aspartic acid	18.37	7.99
Glutamic acid	19.13	5.96
Glycine	9.53	3.72
* Histidine	2.79	0.62
* Isoleucine	7.73	3.34
* Leucine	13.25	5.57
* Lysine	8.92	2.99
* Methionine	1.49	0
* Phenylalanine	7.76	0
Proline	7.94	8.63
Serine	5.78	0.68
* Threonine	5.48	0.29
Tyrosine	2.17	0
* Valine	10.31	4.45
Eta (2-aminoethanol)	0.60	0.57
∑* EAA (%)	35.67	32.22
TAA(g/100 g)	161.93	53.65

TAA: Total amino acids, ∑EAA: The sum of the Essential Amino Acids, * Essential amino acid.

**Table 3 foods-11-01792-t003:** The IC_50_ for ACE-1 inhibition of the 3 kDa permeate generated from *L. digitata*.

Sample Name	IC_50_
Permeate	590 µg/mL

N = 3.

## Data Availability

Data is contained within the article.

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
