# Peer review of "Angiotensin-I-Converting Enzyme Inhibitory Activity of Protein Hydrolysates Generated from the Macroalga Laminaria digitata (Hudson) JV Lamouroux 1813"

_foods, 2022, doi:10.3390/foods11121792_

Round 1
Reviewer 1 Report
Dear Editor
Current study describes Angiotensin-I-converting enzyme inhibitory activity of protein hydrolysates generated from the macroalga Laminaria digitata (Hudson) JV Lamouroux 1813.
1. Introduction must be improved. Most of the references are old. Please replace them with new references
2. The Conclusion is very short. Please improve this section
3. There are several typo errors observed in the main text. For example: HCl not HCL, mL not ml and so one. Please re-check it carefully.
4. Line 112. What did you mean by " nM "?
5. Please support Eqs 1 and 2 with related references.
6. It would be great if you show the process used to generate ACE-1 inhibitory per- 165 meates from L. digitata by graphically scheated instead Figure1.
7. Please compare the obtained results with other similar works
Author Response
Response to reviewer comments
Reviewer 1
Open Review
( ) I would not like to sign my review report
(x) I would like to sign my review report
English language and style
( ) Extensive editing of English language and style required
(x) Moderate English changes required
( ) English language and style are fine/minor spell check required
( ) I don't feel qualified to judge about the English language and style
Yes |
Can be improved |
Must be improved |
Not applicable |
|
Does the introduction provide sufficient background and include all relevant references? |
( ) |
( ) |
(x) |
( ) |
Are all the cited references relevant to the research? |
( ) |
( ) |
(x) |
( ) |
Is the research design appropriate? |
( ) |
(x) |
( ) |
( ) |
Are the methods adequately described? |
(x) |
( ) |
( ) |
( ) |
Are the results clearly presented? |
(x) |
( ) |
( ) |
( ) |
Are the conclusions supported by the results? |
( ) |
( ) |
(x) |
( ) |
Comments and Suggestions for Authors
Dear Editor
Current study describes Angiotensin-I-converting enzyme inhibitory activity of protein hydrolysates generated from the macroalga Laminaria digitata (Hudson) JV Lamouroux 1813.
- Introduction must be improved. Most of the references are old.
2. The Conclusion is very short. Please improve this section:Furthermore, we have added the following references to highlight the novelty of the work:“Chen, J.-C.; Wang, J.; Zheng, B.-D.; Pang, J.; Chen, L.-J.; Lin, H.; Guo, X. Simultaneous Determination of 8 Small Antihypertensive Peptides with Tyrosine at the C-Terminal in Laminaria japonica Hydrolysates by RP-HPLC Method. J. Food Process. Preserv. 2016, 40, 492–501.”The conclusion section now reads as follows: - The following additional references were also added:
- “This work is the first to demonstrate the extraction of proteins and generation of ACE-1 inhibitory hydrolysates from L. digitata using a combination of food grade enzymes. These results build on previous work carried out concerning the health beneficial properties of brown seaweeds linked to polyphenols and carbohydrate fractions, For example, recently Costa and colleagues demonstrated that dietary inclusion of 15% Laminaria digitata, supplemented or not with carbohydrases, could improve the nutritional value of poultry meat without impairing animal growth performance [67]. The authors attributed this to polyphenols and bioactive carbohydrates [67]. To the best of the authors’ knowledge, this is the first paper to report ACE-1 inhibitory activity for protein hydrolysates generated from L. digitata. Previously, Chen et al., reported ACE-1 inhibitory activity from a protein hydrolysate generated from Saccharina (Laminaria) japonica and linked the observed activity to seven bioactive peptides with the amino acid sequences KY, GKY, STKY, AKY, AKYSY, KKFY, KFKY [68]. The next steps in this work will involve characterisation of the bioactive peptides responsible for the observed ACE-1 inhibitory activity. The reported bioactivity of the protein hydrolysates compare favourably with ACE-1 inhibitory peptide hydrolysates identified previously from dairy whey protein hydrolyates (reported ACE IC50 values of 345-1733 µg/ml) [69], meat (beef derived ACE-1 inhibitory peptides had IC50 values of 0.89, 2.69, and 3.09 mM, respectively) [70], mussel Mytilus galloprovincialis (IC50 for the inhibition of ACE of 1.0 to 0.56 mg of protein/mL) [71], microalgal (a few studies have reported IC50 values of microalgal ACE-1 inhibitory peptides ranging from 0.5 to 474.36 μM) [72] and seaweed sources such as Fucus spiralis [73]. L. digitata is cultivated globally through aquaculture and this fact, combined with the potential to extract protein and generate health beneficial ingredients with heart health benefits makes it a suitable candidate for product development. This work adds to the evidence that seaweed-based health maintenance strategies are now considered viable methods to improve global human health.”
- “Costa, M.M.; Pestana, J.M.; Osório, D.; Alfaia, C.M.; Martins, C.F.; Mourato, M.; Gueifão, S.; Rego, A.M.; Coelho, I.; Coelho, D.; et al. Effect of Dietary Laminaria digitata with Carbohydrases on Broiler Production Performance and Meat Quality, Lipid Profile, and Mineral Composition.Animals 2022, 12, 1007. https://doi.org/10.3390/ani12081007” and
- Response: We agree with the reviewer and we have expanded the conclusion section and added the following text and references: “For example, recently Costa and colleagues demonstrated that dietary inclusion of 15% Laminaria digitata, supplemented or not with carbohydrases, could improve the nutritional value of poultry meat without impairing animal growth performance [67]. The authors attributed this to polyphenols and bioactive carbohydrates [67]”. We have also added the following text outlining how this is the first report of ACE-1 inhibitory activity linked to hydrolysate and peptide generation from L. digitata in the following sentences: “To the best of the authors’ knowledge, this is the first paper to report ACE-1 inhibitory activity for protein hydrolysates generated from L. digitata. Previously, Chen et al., reported ACE-1 inhibitory activity from a protein hydrolysate generated from Saccharina (Laminaria) japonica and linked the observed activity to seven bioactive peptides with the amino acid sequences KY, GKY, STKY, AKY, AKYSY, KKFY, KFKY [68]. The next steps in this work will involve characterisation of the bioactive peptides responsible for the observed ACE-1 inhibitory activity. L. digitata is cultivated globally through aquaculture and this fact, combined with the potential to extract protein and generate health beneficial ingredients with heart health benefits makes it a suitable candidate for product development. This work adds to the evidence that seaweed-based health maintenance strategies are now considered viable methods to improve global human health.”
- Response: We would like to thank the reviewer for their comments. We agree with these comments and have included new references in the following locations - in the introduction: on lines: 37-44; lines 48-51, lines 70-77; 48-51.
- Costa, M.M.; Pestana, J.M.; Osório, D.; Alfaia, C.M.; Martins, C.F.; Mourato, M.; Gueifão, S.; Rego, A.M.; Coelho, I.; Coelho, D.; et al. Effect of Dietary Laminaria digitata with Carbohydrases on Broiler Production Performance and Meat Quality, Lipid Profile, and Mineral Composition.Animals 2022, 12, 1007. https://doi.org/10.3390/ani12081007
- Chen, J.-C.; Wang, J.; Zheng, B.-D.; Pang, J.; Chen, L.-J.; Lin, H.; Guo, X. Simultaneous Determination of 8 Small Antihypertensive Peptides with Tyrosine at the C-Terminal in Laminaria japonica Hydrolysates by RP-HPLC Method. J. Food Process. Preserv. 2016, 40, 492–501.
- Pihlanto-Leppälä A, Koskinen P, Piilola K, Tupasela T, Korhonen H. Angiotensin I-converting enzyme inhibitory properties of whey protein digests: concentration and characterization of active peptides. J Dairy Res. 2000 Feb;67(1):53-64. doi: 10.1017/s0022029999003982. PMID: 10717843.
- Choe J, Seol KH, Kim HJ, Hwang JT, Lee M, Jo C. Isolation and identification of angiotensin I-converting enzyme inhibitory peptides derived from thermolysin-injected beef M. longissimus. Asian-Australas J Anim Sci. 2019;32(3):430-436. doi:10.5713/ajas.18.0455.
- Cunha, S.A.; de Castro, R.; Coscueta, E.R.; Pintado, M. Hydrolysate from Mussel Mytilus galloprovincialis Meat: Enzymatic
- Hydrolysis, Optimization and Bioactive Properties. Molecules 2021, 26, 5228. https://doi.org/10.3390/ molecules26175228
- Jiang Q, Chen Q, Zhang T, Liu M, Duan S, Sun X. The Antihypertensive Effects and Potential Molecular Mechanism of Microalgal Angiotensin I-Converting Enzyme Inhibitor-Like Peptides: A Mini Review. Int J Mol Sci. 2021;22(8):4068. Published 2021 Apr 15. doi:10.3390/ijms22084068.
- PAIVA L, LIMA E, NETO A & BAPTISTA J. 2017. Angiotensin I-Converting Enzyme (ACE) Inhibitory Activity, Antioxidant Properties, Phenolic Content and Amino Acid Profiles of Fucus spiralis L. Protein Hydrolysate Fractions. Mar Drugs 15(10): 311. doi: 10.3390/md15100311. 3. There are several typo errors observed in the main text. For example: HCl not HCL, mL not ml and so one. Please re-check it carefully.Response to reviewer comment: We have re-read and edited any spelling mistakes – please see edits to unclean and clean draft of revised paper.
5. Please support Eqs 1 and 2 with related references.Response: We apologise for the omission of equations 1 and 2 regarding ACE-1 inhibition % calculations. We have revised the manuscript to include the equations – please see revision.
- Response: Graph schematic of method replaced the Figure 1, figure as requested and is included in the revised text.
6. It would be great if you show the process used to generate ACE-1 inhibitory per- 165 meates from L. digitata by graphically scheated instead Figure1.- The text now reads: “% ACE-1 inhibition = (Absorbance at 45 nm of blank1 – Absorbance at 450 nm of sample)/(Absorbance at 450 nm of blank1 – Absorbance at 450 nm of blank 2) × 100 [31-32]. Where; blank 1 is the control without the addition of any inhibitor, blank 2 is the reagent blank, and the inhibitor is the positive control (Captopril ©) or test sample (protein hydrolysate fraction).”
- Response: We have clarified that we meant nanomolar when using nM on line 112.
4. Line 112. What did you mean by " nM "?
Please compare the obtained results with other similar works
Response: We thank the reviewer for these comments. We have compared the results obtained in terms of ACE IC50 values in the revised conclusion section as outlined in response earlier. We have highlighted work carried out with Laminaria japonica previously also.

Reviewer 2 Report
Title: Angiotensin-I-converting enzyme inhibitory activity of protein hydrolysates generated from the macroalga Laminaria digitata (Hudson) JV Lamouroux 1813
- The work has several errors overall. All scientific terms should be italicized. Please adhere to the format provided in the guideline and proofread the work again.
- The article, in terms of the goals and importance of the topic, is written scattered. It needs to be rewritten consistently and interestingly.
- The importance of the issue and the need to do so are not properly stated. Please provide your gap of knowledge and research question properly. Why is this work so important?
- In Table 3, which contains the most important data, I recommend that the authors provide the results using the IC50 derived from the 3kDa permeate in comparison to the drug control. The percent inhibition may be utilized, although it is difficult to compare to the others.
- The discussion is inappropriate. The authors simply compared their work to earlier ones. Please explain how and why your findings are superior than those of others.
- Please improve quality and style of Fig 1, 3 and 4.
- Is there any information on the heavy metal content of the seaweeds used? Is your extraction process capable of removing heavy metals? Heavy metal pollution is one of the bottlenecks in seaweed industries, thus it's worth mentioning in the discussion.
Author Response
Open Review
(x) I would not like to sign my review report
( ) I would like to sign my review report
English language and style
( ) Extensive editing of English language and style required
(x) Moderate English changes required
( ) English language and style are fine/minor spell check required
( ) I don't feel qualified to judge about the English language and style
Yes |
Can be improved |
Must be improved |
Not applicable |
|
Does the introduction provide sufficient background and include all relevant references? |
( ) |
( ) |
(x) |
( )** added additional refs Diane did#[55-58] |
Are all the cited references relevant to the research? |
(x) |
( ) |
( ) |
( ) |
Is the research design appropriate? |
( ) |
(x) |
( ) |
( ) |
Are the methods adequately described? |
(x) |
( ) |
( ) |
( ) |
Are the results clearly presented? |
( ) |
( ) |
(x) |
|
Are the conclusions supported by the results? |
( ) |
(x) |
( ) |
( ) |
Comments and Suggestions for Authors
Title: Angiotensin-I-converting enzyme inhibitory activity of protein hydrolysates generated from the macroalga Laminaria digitata (Hudson) JV Lamouroux 1813
- The work has several errors overall. All scientific terms should be italicized. Please adhere to the format provided in the guideline and proofread the work again.
Response: Spellcheck and all checked for scientific terms, and proofread, SI units checked and corrected.
- The article, in terms of the goals and importance of the topic, is written scattered. It needs to be rewritten consistently and interestingly. ?
Response: Importance of topic and context supplied in lines: 35-77. We hope this is now acceptable.
- The importance of the issue and the need to do so are not properly stated. Please provide your gap of knowledge and research question properly. Why is this work so important?
Response: The Gap in knowledge addressed, importance of work (addressed in point 2 above) and research question in Lines: 70-77.
- In Table 3, which contains the most important data, I recommend that the authors provide the results using the IC50 derived from the 3kDa permeate in comparison to the drug control. The percent inhibition may be utilized, although it is difficult to compare to the others.
Response: We agree with the reviewer and we have edited this significantly in the revised manuscript to reflect the requirements of reviewer 2.
- The discussion is inappropriate. The authors simply compared their work to earlier ones. Please explain how and why your findings are superior than those of others?
Response: We have added details on novelty of this work, the potential application and product development opportunity from the results from this work, and where this bioactivity sits in reference to other brown seaweeds: lines 319-380. The novelty lies in the fact that peptide hydrolysates from L. digitata with ACE-1 inhibitory activities were not previously identified. In addition, the percentage increase in protein concentration in the permeate fraction is highlighted.
- Please improve quality and style of Fig 1, 3 and 4.
Response: As requested Figure 1, has been improved and converted to the schematic (Reviewer one comment), and Figure 3 & 4 have been improved, re-imaged and put together to clarify the information from both these figures, so the % yield change is included, through each extracted fraction.
- Is there any information on the heavy metal content of the seaweeds used? Is your extraction process capable of removing heavy metals? Heavy metal pollution is one of the bottlenecks in seaweed industries, thus it's worth mentioning in the discussion. ( L. dig feed grade, supplied by company for feed grade use).
Response: The focus of this work was not heavy metal determination. However, the material used and supplied by the company is considered feed grade and therefore is compliant with regulations within the EU concerning what is allowed in seaweeds. We have added a sentence on this in the discussion section and conclusion. The text now reads as follows: “One concern about seaweed consumption is exposure to heavy metals such as arsenic, aluminum, cadmium, lead, rubidium, silicon, strontium, and tin. One concern about seaweed consumption is exposure to heavy metals such as arsenic, aluminum, cadmium, lead, rubidium, silicon, strontium, and tin. In this work, seaweed was supplied from a feed and food company and the raw material was compliant with EU legislation [73].”

Round 2
Reviewer 2 Report
11. Page 1, line 38: pharma or pharmaceutical?
22. I think the authors have a misunderstanding concerning the ACE-1 data. It is improper to present Table 3 in this manner. Table 3 should use the IC50 value rather than the percentage inhibition. An IC50 can be used by all readers to evaluate the efficiency of your sample to that of others. When concentration is not present, the percent inhibition cannot be used to compare efficiency. You cannot report percent inhibition alone; you must report it in conjunction with extract concentration. For example, can you use page 9 line 312-313? and page 10 line 352-356? to compare with your data or others when concentration is missing?
33. The authors also reported wrong information in the abstract/result section. You said that Captopril had an IC50 value of 500 μg/mL in the abstract. However, page 9 line 307-308, you said that Captopril a drug used to control AHT that achieved 100 % inhibition when assayed at a concentration of 500 μg/mL. Therefore, which one is correct?
Author Response
Comments and Suggestions for Authors
***Additional Revisions to Introduction” Lines: 72-78 Sargassum siliquosum and Sargassum polycystum previously displayed ACE-1 IC50 values of 0.03–0.42 mg/ml, at concentrations ranging from 0.2–1.0 mg/ml [58]. Phlorotannins from Sargassum wightii have also shown ACE-1 inhibitory activity, with a reported IC50 value of 56.96 µg/ml, which equates to the IC50 value previously reported for Captopril © of 51.79 µg/ml [57]. In addition, protein hydrolysates extracted from the green seaweed Ulva intestinalis had ACE-1 inhibitory IC50 values of 0.183 mg/mL and 0.179 mg/mL when assessed previously [45]”.
- 1 Page 1, line 38: pharma or pharmaceutical?
Response: we thank the reviewer for their comments: and corrections have been applied to: line 38 has been corrected to read pharmaceuticals
- I think the authors have a misunderstanding concerning the ACE-1 data. It is improper to present Table 3 in this manner. Table 3 should use the IC50 value rather than the percentage inhibition. An IC50 can be used by all readers to evaluate the efficiency of your sample to that of others. When concentration is not present, the percent inhibition cannot be used to compare efficiency. You cannot report percent inhibition alone; you must report it in conjunction with extract concentration. For example, can you use page 9 line 312-313? and page 10 line 352-356? to compare with your data or others when concentration is missing?
Response: we thank the reviewer for their comments. This section now reads as:
The ACE-1 IC50 value obtained for the L. digitata 3-kDa permeate was 590 µg/mL. This is a promising result when compared with Captopril© a drug used to control AHT that achieved 100 % inhibition, and an IC50 of 500 µg/mL (Table 3). In addition, the IC50 value of a protein hydrolysate generated from Undaria sp. (Wakame) was shown as 86 µg/ml. This hydrolysate was tested further for antihypertensive activity in an animal trial, where it was orally administered to spontaneously hypertensive rats (SHRs). The hydrolysate was fed orally at concentrations of 10 mg and 100 mg protein/kg body weight (BW), and compared to the positive antihypertensive drug Captopril© which was administered at a dose of 5 mg/kg BW. Three hours post administration of 10 mg/kg BW of the hydrolysate, systolic blood pressure (SBP) dropped by 19.3 ± 3.1 mm Hg (p<0.01) and after six hours post administration of 100 mg protein/kg BW of the hydrolysate, SBP decreased by 23.5 ± 5.3 mm Hg (p<0.05). These in vivo values compared favourably to Captopril©, which decreased SBP in SHRs by 30.6 ± 10.2 mm Hg (p < 0.05), 3 hr post administration [48]. In addition, previously generated hydrolysates from the green seaweed, Ulva intestinalis reported ACE-1 inhibition values of 0.183 mg/mL and 0.179 mg/mL when assayed at different concentrations. Observed ACE-1 inhibition was attributed to interaction of Zn2+ and the two novel peptides identified and characterized from the hydrolysate [45].”
- The authors also reported wrong information in the abstract/result section. You said that Captopril had an IC50 value of 500 μg/mL in the abstract. However, page 9 line 307-308, you said that Captopril a drug used to control AHT that achieved 100 % inhibition when assayed at a concentration of 500 μg/mL. Therefore, which one is correct?
Response: we thank the reviewer for their comments and this observation which was an error on our part. The data within the abstract section is correct, and we revised the results section.
(Abstract: lines 27-28) now reads “The concentration of permeate that resulted in 75% inhibition of ACE-1 (the IC50 value) was 590µg/ml, which compares well with Captopril© that resulted in 100% inhibition of ACE-1, at an IC50 value of 500 µg/ml.”
Submission Date
10 April 2022
Date of this review
28 May 2022 13:50:05
